# Multi-label Learning with Random Circular Vectors

## Abstract

The extreme multi-label classification (XMC) task involves learning a classifier that can predict from a large label set the most relevant subset of labels for a data instance. While deep neural networks (DNNs) have demonstrated remarkable success in XMC problems, the task is still challenging because it must deal with a large number of output labels, which make the DNN training computationally expensive. This paper addresses the issue by exploring the use of random circular vectors, where each vector component is represented as a complex amplitude. In our framework, we can develop an output layer and loss function of DNNs for XMC by representing the final output layer as a fully connected layer that directly predicts a low-dimensional circular vector encoding a set of labels for a data instance. We conducted experiments on synthetic datasets to verify that circular vectors have better label encoding capacity and retrieval ability than normal real-valued vectors. Then, we conducted experiments on actual XMC datasets and found that these appealing properties of circular vectors contribute to significant improvements in task performance compared with a previous model using random real-valued vectors, while reducing the size of the output layers by up to 99%.

## 1 Introduction

Extreme multi-label classification (XMC) problems arise in various domains, such as product recommendation systems (Jain et al., 2016), labeling large encyclopedia (Dekel & Shamir, 2010; Partalas et al., 2015), instance-level image recognition (Deng et al., 2010; Ridnik et al., 2021) and natural language generation (Mikolov et al., 2013; Li et al., 2022). The XMC task involves learning a classifier which can predict from a large label set the most relevant subset of labels for a data instance. Recent work has focused on deep neural network (DNN) models (Liu et al., 2017; You et al., 2019; Chang et al., 2020; Zhang et al., 2021; Dahiya et al., 2023; Jain et al., 2023) that deliver task performances superior to those of early approaches using linear predictors (Babbar & Schölkopf, 2017; Prabhu et al., 2018).

While DNN models have brought great performance improvements, the XMC task still remains a challenge mainly due to the extremely large output space. Since a large number of output labels make it difficult to train DNN models efficiently, various methods for improving training efficiency have been proposed (Khandagale et al., 2020; Wydmuch et al., 2018; Jiang et al., 2021; Ganesan et al., 2021). Among the previous studies, Ganesan et al. (2021) presented a promising method that employs random real-valued vectors for reducing the output layer size of DNN models. In this approach, a high-dimensional output space vector is replaced with a low-dimensional random vector encoding the relevant label information for a data instance. Then, DNN models are trained to predict the label-encoded vector directly. After the model generates a vector, it can be checked approximately whether a label is encoded in it or not through a vector comparison using the cosine similarity between the output vector and a vector that the label is assigned to. The basic idea of the label encoding and retrieval framework relies on the theory of Holographic Reduced Representations (Plate, 1995), which was developed in the cognitive neuroscience field.

However, random real-valued vectors do not have sufficient ability for representing data instances that belong to many class concepts. As our experiments in § 3 show, the label retrieval accuracy decreases markedly as the number of class labels encoded in a vector increases. To alleviate the issue, this paper presents a novel method that uses *circular* vectors instead of real-valued vectors.

Each element of a circular vector takes a complex amplitude as its value; i.e., the vector element is represented by an angle ranging from $-\pi$ to $\pi$. Since an angle can be represented by a real value, the memory cost for the circular vector representation is the same as that for a normal real-valued vector. In spite of this fact, surprisingly, circular vectors have better label encoding and retrieval capacities than real-valued vectors. One of the challenges in applying circular vectors to DNN models is how to adapt the output layer to a circular vector. In § 4, we describe our neural network architecture that uses circular vectors in the output layer. Our experimental results on XMC datasets show that our method based on circular vectors significantly outperforms a previous model using real-valued vectors, while reducing the size of the output layers by up to 99%.

## 2 Previous Study: Learning with Holographic Reduced Representations

Several vector symbolic architectures have been developed in the field of cognitive neuroscience, including Tensor Product Representations (Smolensky, 1990), Binary Spatter Code (Kanerva, 1996), Binary Sparse Distributed Representations (Rachkovskij, 2001), Multiply-Add-Permute (Gayler, 2004), and Holographic Reduced Representations (HRR) (Plate, 1995). Among them, HRR is a successful architecture for distributed representations of compositional structures. To model complex structured prediction tasks in a vector space that involve key-value stores, sequences, trees and graphs, many prior studies have explored how to use HRR in various machine learning frameworks; Recurrent Neural Networks (Plate, 1992), Tree Kernels (Zanzotto & Dell'Arciprete, 2012), Knowledge Graph Representation Learning (Nickel et al., 2016), Long-short Term Memory Networks (Danihelka et al., 2016), Transformer Networks (Alam et al., 2023), and among others. In particular, Ganesan et al. (2021) presented a general framework based on the HRR architecture for efficient multi-label learning of DNN models. To clarify the motivation of our study, we will review the framework in more detail in the following subsections.

### 2.1 Holographic Reduced Representations (HRR)

In the HRR architecture, terms in a domain are represented by real-valued vectors. Here, we assume that each vector is independently sampled from a Gaussian distribution $\mathcal{N}(0, \mathbf{I}_d \cdot d^{-1})$, where $d$ is the vector dimension size and $\mathbf{I}_d$ is the $d \times d$ identity matrix. To bind an association of two terms represented by vectors $\mathbf{a}$ and $\mathbf{b}$, respectively, HRR uses circular convolution, denoted by the mathematical symbol $\otimes$:

$$\mathbf{a} \otimes \mathbf{b} = \mathcal{F}^{-1}(\mathcal{F}(\mathbf{a}) \odot \mathcal{F}(\mathbf{b})) \tag{1}$$

where $\odot$ is element-wise vector multiplication. Note that the circular convolution can be computed by using a fast Fourier transform (FFT) $\mathcal{F}$ and inverse FFT $\mathcal{F}^{-1}$, but they require $\mathcal{O}(d \log d)$ computation time. Given several associations $\mathbf{a} \otimes \mathbf{b}$, $\mathbf{c} \otimes \mathbf{d}$ and $\mathbf{e} \otimes \mathbf{f}$, the vectors can be superposed to represent their combination: $\mathbf{S} = (\mathbf{a} \otimes \mathbf{b}) \oplus (\mathbf{c} \otimes \mathbf{d}) \oplus (\mathbf{e} \otimes \mathbf{f})$, where the "superposition" operator $\oplus$ is just normal vector addition $+$. The HRR architecture also provides the inversion operation $\dagger$:

$$\mathbf{a}^\dagger = \mathcal{F}^{-1}(\frac{1}{\mathcal{F}(\mathbf{a})}). \tag{2}$$

The inversion operation can be used to perform "unbinding". For an example, it allows the reconstruction of a noisy version of $\mathbf{d}$ to be recreated from the memory $\mathbf{S}$ and a cue $\mathbf{c}$: $\mathbf{S} \otimes \mathbf{c}^\dagger \approx \mathbf{d}$. Finally, the "similarity" operation is defined as the dot-product $\mathbf{a}^\mathrm{T}\mathbf{b}$. Using the similarity operation, we can check approximately whether $\mathbf{a}$ exists in a memory $\mathbf{S}$ if $\mathbf{S}^\mathrm{T}\mathbf{a} \approx 1$ or not present if $\mathbf{S}^\mathrm{T}\mathbf{a} \approx 0$.

### 2.2 Multi-label Learning with HRR

Ganesan et al. (2021) introduced a novel method using HRR for reducing the computational complexity of training DNNs for XMC tasks. Let $L$ be the number of class labels in an XMC task. The basic idea behind the approach of (Ganesan et al., 2021) is quite intuitive; for efficient DNN training, an $L$-dimensional output (teacher) vector is replaced with a $d$-dimensional real-valued vector encoding the relevant label information for a data instance. By assuming $d \ll L$, we can dramatically reduce the output layer size of the DNN model.

In this approach, each class label $y$ is assigned to a $d$-dimensional vector $\mathbf{c}_y \in \mathbb{R}^d$. Then, the label information for a data instance $x$ is represented as a *label vector* $\mathbf{S}_x \in \mathbb{R}^d$:

$$\mathbf{S}_x = \bigoplus_{p \in \mathcal{Y}_x} \mathbf{p} \otimes \mathbf{c}_p \tag{3}$$

where $\mathcal{Y}_x$ denotes the set of class labels that $x$ belongs to and $\mathbf{p} \in \mathbb{R}^d$ represents the positive class concept.[1] To train a DNN model $f(\mathbf{x})$ that generates $\hat{\mathbf{S}}_x \in \mathbb{R}^d \approx \mathbf{S}_x$, Ganesan et al. (2021) define a loss function:

$$loss = \sum_{p \in \mathcal{Y}_x} (1 - sim((\hat{\mathbf{S}}_x \otimes \mathbf{p}^\dagger), \mathbf{c}_p)). \tag{4}$$

To prevent the model from maximizing the magnitudes of the output vectors, Ganesan et al. (2021) used the cosine similarity as $sim(\cdot, \cdot)$, which is a normalized version of the dot product that ranges from -1 to 1. In the inference phase, labels can also be ranked according to the cosine similarity computed by $sim(\hat{\mathbf{S}}_x \otimes \mathbf{p}^\dagger, \mathbf{c}_p)$ for each label $p$. Moreover, Ganesan et al. (2021) introduced a novel vector *projection* method to reduce the effect of the variance of the similarity computation:

$$\pi(\mathbf{x}) = \mathcal{F}^{-1} \left( \dots, \frac{\mathcal{F}(\mathbf{x})_j}{|\mathcal{F}(\mathbf{x})_j|}, \dots \right). \tag{5}$$

Here, each HRR vector $\mathbf{x}$ is initialized with $\mathbf{x} \stackrel{\mathrm{d}}{=} \pi\left(\mathcal{N}(0, \mathbf{I}_d \cdot d^{-1})\right)$, which ensures each element of the vector in the frequency domain is unitary; i.e., the complex magnitude is one.

## 3 MULTI-LABEL REPRESENTATIONS WITH CIRCULAR VECTORS

In this section, we show through experiments that random real-valued vectors actually do not have sufficient ability for representing data instances that belong to many classes. The reason is mainly due to the projection operation in Equation 5. As described in § 2, the projection operation was proposed as a way to reduce the effect of the variance of the similarity computation, but each element of the superposition between two normalized vectors via the projection is no longer unitary. Thus, the effect of the projection decreases when a label vector encodes more class labels. To alleviate the issue, we developed a simple alternative that forces all vector elements to be unitary in the complex domain even after the superposition operation. We describe the details in the following subsection.

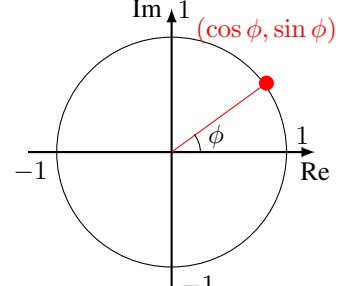

Figure 1: The unit circle in the complex plane with coordinates. The angle $\phi$ represents an element of the circular vector $\bar{\phi}$.

### 3.1 HRR WITH CIRCULAR VECTOR

Our idea is to use *circular* vectors instead of real-valued vectors. Circular vectors have a complex amplitude (see Figure 1), which can be represented by a real value $\phi$ ranging from $-\pi$ to $\pi$. However, to force all vector elements to be unitary after any operations, we require a special HRR system for circular vectors. In this paper, we borrow the concept of a circular HRR (CHRR) system from (Plate, 2003).

Table 1 compares the HRR operations of the standard and circular systems. For circular vectors, each element must be sampled from a uniform distribution $\mathcal{U}(-\pi, \pi)$ over $(-\pi, \pi]$. The binding $\otimes$ and inversion $\dagger$ of CHRR are implemented with the standard vector arithmetic operations like addition and subtraction. The similarity of two circular vectors can be simply determined from the sum of the cosines of the differences between angles. On the other hand, superposition is somewhat tricky because in general the sum of unitary complex values does not lie on the unit circle. For each pair of elements $\phi_j$ and $\theta_j$ of two circular vectors $\bar{\phi}$ and $\bar{\theta}$, the result of superposition is $\angle(e^{i \cdot \phi_j} + e^{i \cdot \theta_j})$. Here, $\angle(v)$ extracts an angle of a complex value $v$ and discards the magnitude of $v$. Since all of these

---

[1]We can encode information on negative labels into a label vector as well as positive ones, but as shown in (Ganesan et al., 2021), the negative label information does not contribute to improving XMC task performance. Thus, in this paper, we will omit discussion on negative labels for notational brevity.

Table 1: Comparison of HRR operations on real-valued and circular vectors

| Operation | Real-valued (Ganesan et al., 2021) | Circular |
|---|---|---|
| vector | $\mathbf{x} = [x_0, \ldots, x_{d-1}]$ | $\bar{\phi} = [\phi_0, \ldots, \phi_{d-1}]$ |
| random vector | $\mathbf{x} \stackrel{\mathrm{d}}{=} \pi\left(\mathcal{N}(0, \mathbf{I}_d \cdot d^{-1})\right)$ | $\phi_j \stackrel{\mathrm{d}}{=} \mathcal{U}(-\pi, \pi)$ |
| binding | $\mathbf{x} \otimes \mathbf{y} = \mathcal{F}^{-1}(\mathcal{F}(\mathbf{x}) \odot \mathcal{F}(\mathbf{y}))$ | $\bar{\phi} \otimes \bar{\theta} = [(\phi_0 + \theta_0) \bmod 2\pi, \ldots,$ $(\phi_{d-1} + \theta_{d-1}) \bmod 2\pi]$ |
| unbinding | $\mathbf{x} \otimes \mathbf{y}^\dagger = \mathbf{x} \otimes \mathcal{F}^{-1}(\frac{1}{\mathcal{F}(\mathbf{y})})$ | $\bar{\phi} \otimes \bar{\theta}^\dagger = -\bar{\theta} \otimes \bar{\phi}$ |
| similarity | $sim(\mathbf{x}, \mathbf{y}) = \mathbf{x}^{\mathrm{T}} \mathbf{y}$ | $sim(\bar{\phi}, \bar{\theta}) = \frac{1}{d} \sum_j \cos(\phi_j - \theta_j)$ |
| superposition | $\mathbf{x} \oplus \mathbf{y} = \mathbf{x} + \mathbf{y}$ | $\bar{\phi} \oplus \bar{\theta} = [\angle(e^{i \cdot \phi_0} + e^{i \cdot \theta_0}), \ldots,$ $\angle(e^{i \cdot \phi_{d-1}} + e^{i \cdot \theta_{d-1}})]$ |

(a) HRR(w/Proj)

(b) CHRR

Figure 2: Retrieval accuracies of HRR(w/Proj) and CHRR. The number of dimensions $d$ was $1, \ldots, 1024$ and the number of positive classes $k$ was $1, \ldots, 50$.

operations do not affect the unitary property of circular vectors, we no longer need the projection normalization process. Our framework also has an advantage in computational cost; we can avoid the FFT and inverse FFT operations, which take $\mathcal{O}(d \log d)$ computation time.

## 3.2 RETRIEVAL ACCURACY EXPERIMENT

We experimentally demonstrated CHRR's capacity by comparing its retrieval accuracy with that of HRR. The experiment attempted to verify how accurately the positive class vector can be retrieved from a memory vector. For a data instance $x$, let $\mathbf{c}_p$ be a vector for a positive class $p$ to which $x$ belongs, and let $\mathbf{p}$ be a vector for the positive class concept label. The binding and superposition operations allow us to represent all positive classes for $x$ as $\mathbf{R}$:

$$\mathbf{R} = \bigoplus_{p \in \mathcal{Y}_x} (\mathbf{p} \otimes \mathbf{c}_p). \tag{6}$$

In the experiment, we generated a database consisting of $N = 1,000$ random $d$-dimensional vectors ($\mathbf{c}_j \in \mathbb{R}^d$, for all $j \in [1, \ldots, N]$). Then, to create $\mathbf{R}$, we randomly selected $k$ vectors from the database to be $\mathbf{c}_p$ and one vector to be $\mathbf{p}$. As shown in Equation 6, the $k$ associations can be superposed to represent $\mathbf{R}$. To retrieve $\mathbf{c}_p$ from $\mathbf{R}$, we used the unbinding operation to decode a noisy version of the vector $\mathbf{c}_p$ from $\mathbf{R}$, as $\hat{\mathbf{c}}_p = \mathbf{R} \otimes \mathbf{p}^\dagger$. For each $j \in [1, \ldots, N]$, we computed the similarity $s_j = sim(\hat{\mathbf{c}}_p, \mathbf{c}_j)$ between the decoded vector $\hat{\mathbf{c}}_p$ and the individual vector $\mathbf{c}_j$. After that, we compiled the top-$k$ label list according to the similarity scores $s_j$. To evaluate the retrieval accuracy, we measured the percentage of class labels in the list, whose vectors were encoded into the memory $\mathbf{R}$. By varying the number of dimensions $d = 1, \ldots, 1024$ and the number of binding pairs $k = 1, \ldots, 50$, we plotted the accuracies as a heat-map (Figure 2, where warmer colors indicate higher accuracy).[2] The results clearly show that CHRR has better retrieval accuracies than those

---

[2]Schlegel et al. (2021) also demonstrated that CHRR has a higher retrieval capacity compared with HRR. Yet, they used all distinct vectors: $\mathbf{R} = (\mathbf{a} \otimes \mathbf{b}) \oplus (\mathbf{c} \otimes \mathbf{d})$, and did not use a fixed $\mathbf{p}$: $\mathbf{R} = (\mathbf{p} \otimes \mathbf{a}) \oplus (\mathbf{p} \otimes \mathbf{b})$. Therefore, we changed their experimental settings to fit the XMC learning with HRR.

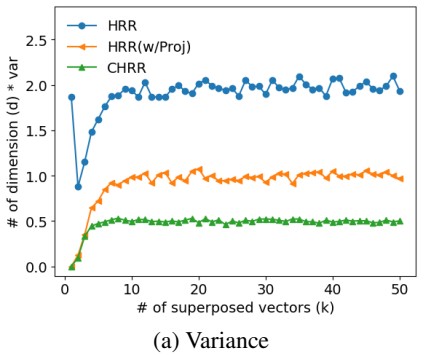 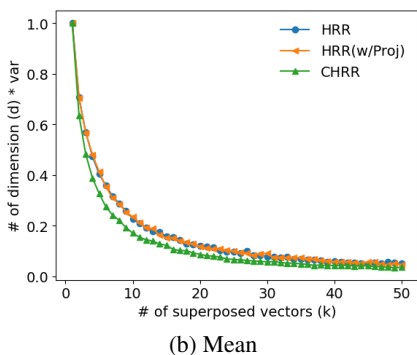

(a) Variance

(b) Mean

Figure 3: Variance and mean of the similarities of HRR, HRR(w/Proj), and CHRR. We fixed the number of dimensions $d$ to $400$ and varied the number of positive classes $k$ to from $1$ to $50$.

of HRR. Moreover, the larger the number of superposed vectors ($k$) is, the bigger the performance difference between CHRR and HRR becomes. Hence, this tendency indicates that CHRR is more suitable than HRR for encoding many labels.

### 3.3 Variance Comparison Experiment

In § 3.2, we confirmed that CHRR exhibits superior retrieval ability to HRR. There is a possibility that the CHRR's similarity operation reduces the variance more than the projection does. The experiment reported below was conducted to check the numerical stability of the CHRR's similarity operation. To create $\mathbf{R}$ as Equation 6, we generated $k$ random vectors $\mathbf{c}_p$ and $\mathbf{p}$. We extracted a noisy version of $\mathbf{c}_p$ from $\mathbf{R}$ as $\hat{\mathbf{c}}_p = \mathbf{R} \otimes \mathbf{p}^{\dagger}$. For each $j \in [1, \ldots, k]$, we measure the similarity between $\hat{\mathbf{c}}_\mathbf{p}$ and $\mathbf{c}_j$ as $s_j = sim(\hat{\mathbf{c}}_p, \mathbf{c}_j)$. We plotted the variances and means of the similarities in Figure 3 (a) and (b), respectively. We fixed the number of dimensions $d$ to $400$ and varied the number of binding pairs $k = 1, \ldots, 50$. Our experiments compared three methods, CHRR, HRR proposed in (Plate, 1995), and HRR with the projection of (Ganesan et al., 2021) (HRR(w/Proj)).

Figure 3 (a) shows that as $k$ increases, the variances of all methods tend to converge. However, while the variance converges, the mean also decreases near zero, as shown in Figure 3 (b). Therefore, as the number of superposed vectors $k$ increases, the impact of variance becomes relatively larger. Regarding the variance, we can see the need for the projection, since the HRR(w/Proj) is more suppressed than the original HRR. Yet, we found that CHRR is most suppressed; that is, CHRR is more numerically stable than HRR(w/Proj). As for the mean, the three methods had roughly comparable performances. Although the mean approached zero as $k$ increased, this is not a problem in using similarity for compiling a ranking list of labels.

## 4 Neural network architecture

One of the challenges in adapting CHRR to XMC tasks is how to adapt the output layer of DNN models to a circular vector because it has a cyclic feature; i.e., $\theta = 2\pi n \times \theta$, where $n \in \mathbb{Z}$. To meet it, we developed a neural network for predicting angles that considers the cyclic feature during the training. The key idea was to represent the output in Cartesian coordinates, which can uniquely represent a point on a unit circle. Then, we converted the output into polar coordinates to obtain angles.

### 4.1 Architecture for circular vector

We used fully connected (FC) networks in all of the experiments. They were each composed of a $F$-dimensional input layer, two $h$-dimensional hidden layers with ReLU activation (Agarap, 2018), and a $d'$-dimensional output layer. That is, they had the same architecture except for the output layer.

We selected two baselines from Ganesan et al. (2021) by using the FC networks. The first baseline had $L$ output nodes and each node is used to binary classification (we refer to it below as FC). The second baseline was the method using HRR as described in § 2.2. It had $d$ output nodes (we refer to it below as HRR).

Our network for CHRR represented a pair of the outputs as a point on a unit circle on Cartesian coordinates; i.e., $(\cos\phi, \sin\phi)$, as shown in Figure 1. Then we converted the point into polar coordinates $(1, \phi)$, and used $\phi$ as an element of the predicted label vector. Let $\hat{s} \in \mathbb{R}^{2d}$ be the raw output vector, and $\hat{S} \in \mathbb{C}^d$ be the converted circular vector. We represented $d$ pairs from $\hat{s}$ in Cartesian coordinates as $a_i = (x_i, y_i)$. Then, we normalized them to satisfy $\|a_i\| = 1$. Although there was a similar work for an angle prediction using a neural network (Heffernan et al., 2015), they used $\arctan\frac{y}{x}$ for the conversion whose range was limited to $\left[\frac{-\pi}{2}, \frac{\pi}{2}\right]$. Instead, we used the atan2 function (Organick, 1966), which can convert a $(x, y)$ point to a corresponding angle $(-\pi, \pi]$. Finally, we adapted the atan2 to $a_i$ to obtain $\hat{S}_i$. We named this method as CHRR.

## 4.2 IMPACT OF MODEL ARCHITECTURE

Because the number of the output nodes of CHRR $(2d)$ is twice as that of HRR $(d)$, the total model size of CHRR also increases. Therefore, we conducted two different experiments using the same model size as HRR (see § 5.4 for the results). The first experiment changed the network architecture of CHRR. Figure 4 compares the architectures of CHRR and the changed model (CHRR-Half). We made CHRR-Half by splitting the second hidden layer's nodes and output nodes of CHRR in half. This resulted in two sets of $\frac{h}{2}$ hidden nodes and $d$ output nodes. Then we connected one set of hidden nodes to one set of output nodes, and the other set of hidden nodes to the other set of output nodes. As a result, $2 \times (\frac{h}{2} \times d) = h \times d$ parameters were obtained, which equals the number of parameters between the second hidden layer and the output layer in HRR. The results of the experiment in § 5.4 showed no significant difference in performance between CHRR and this model. Therefore, the increase in the model size of CHRR is not a big issue.

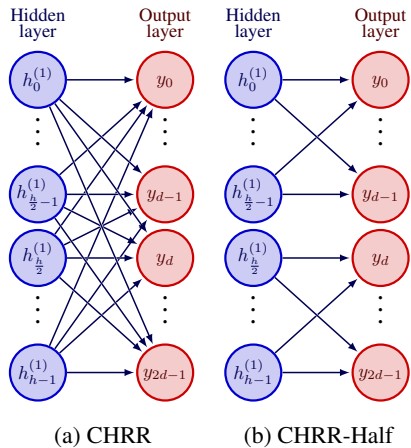

(a) CHRR     (b) CHRR-Half

Figure 4: Comparison of (a) CHRR and (b) CHRR-Half architectures

In the second experiment, to demonstrate the advantage of the proposed architecture against naive implementation, we used the same network architecture as HRR, and mapped the real-valued outputs to angles with activation functions. We tried two activation functions, $\sin$ and $\tanh$ to map the outputs to $[-1, 1]$; then the output was multiplied by $\pi$ to obtain $(-\pi, \pi]$ outputs. We named these models as CHRR-sin and CHRR-tanh. Both showed more modest levels of performance compared with CHRR.

## 5 EXPERIMENT ON XMC DATASETS

To examine the advantages of circular vectors, we conducted experiments on several XMC datasets. Note that achieving the state-of-the-art performance on XMC datasets was not the goal of this study, which focuses on the efficiency of the learning method with circular vectors. Therefore, we compared our method with FC networks and HRR, but it is also a fact that FC model is a simple but strong baseline for the XMC task (see (Ganesan et al., 2021)).

### 5.1 DATASETS

We evaluated our method on the four datasets for text XMC tasks from Bhatia et al. (2016). Table 2 shows the details of the datasets. The features for each sample is a bag-of-words of $F$ words. These datasets have a large number of labels $(L)$, up to 205,443. In addition, the maximum of the average number of labels per samples $(\hat{L})$ is 75.74 of Delicious-200K.

### 5.2 EVALUATION METRICS

We evaluated each method by using precision at $k$ (P@$k$) and the propensity score at $k$ (PSP@$k$), which are commonly used metrics in the XMC task. P@$k$ is the proportion of true labels in the

Table 2: Details of the datasets from Bhatia et al. (2016). Here, $N_{train}$ is the number of training samples, $N_{test}$ is the number of test samples, $F$ is the number of the dimensions of bag-of-words features, $L$ is the number of labels, $\bar{L}$ is the average number of samples per label, and $\hat{L}$ is the average number of labels per sample.

| Dataset | $N_{train}$ | $N_{test}$ | $F$ | $L$ | $\bar{L}$ | $\hat{L}$ |
|---|---|---|---|---|---|---|
| Delicious | 12,920 | 3,185 | 500 | 983 | 311.61 | 19.03 |
| EURLex-4K | 15,539 | 3,809 | 5,000 | 3,993 | 25.73 | 5.31 |
| Wiki10-31K | 14,146 | 6,616 | 30,938 | 101,938 | 8.52 | 18.64 |
| Delicious-200K | 196,606 | 100,095 | 782,585 | 205,443 | 2.29 | 75.74 |

top-$k$ predictions (Equation 7). PSP@$k$ is a variation of precision that takes into account the relative frequency of each label (Equation 8).

$$\text{P@}k = \frac{1}{k} \sum_{l \in \text{rank}_k(\hat{\mathbf{y}})} \mathbf{y}_l \qquad (7) \qquad \text{PSP@}k = \frac{1}{k} \sum_{l \in \text{rank}_k(\hat{\mathbf{y}})} \frac{\mathbf{y}_l}{p_l} \qquad (8)$$

where $\text{rank}_k(\hat{\mathbf{y}})$ is the ranking of all labels in the predicted $\hat{\mathbf{y}}$ and $p_l$ is the relative frequency of the $l$-th label. We used $k = 1, 5, 10, 20$ for P@$k$, and $k = 1, 5, 10, 20$ for PSP@$k$ in the experiments described below.

### 5.3 EXPERIMENTAL SETTINGS

We evaluated three models (FC, HRR, CHRR) on the four datasets. For the implementation of FC and HRR, we used the scripts provided by Ganesan et al. (2021) available at the GitHub URL.[3] We implemented CHRR by using PyTorch (Paszke et al., 2019). The training methods and the model architectures basically followed the scripts provided by Ganesan et al. (2021). In CHRR, we varied the dimension of the symbol vectors ($d$) $\{100, 400, 800, 1000\}$. To investigate the possibility that a larger hidden layer size $h$ improves the learning effect in FCs with large output dimensionality, we conducted experiments with three settings of hidden layer size ($h$) $\{768, 1024, 2048\}$. For main results, we chose $d = 800$ and $h = 768$ for CHRR and $h = 2048$ for FC. All experiments are conducted with two hidden layers. In Appendix A, we also report the performance impact of the number of hidden layers. Because the performance of neural networks depends on the initial state, we took the average scores of five trials as the evaluation result except for Delicious-200K [4].

### 5.4 RESULTS AND DISCUSSION

Table 3 lists P@1, P@5, P@10, PSP@1, PSP@5, and PSP@10 of FC and CHRR on each dataset. CHRR achieves up to 99% output dimensional compression while providing performance comparable to FC. In particular, the performance difference between CHRR and FC is seen in Delicious-200K. This means that CHRR is effective for such datasets with a large number of labels per sample. Figure 5 shows the impact of the dimensionality size $d$ of the HRR and CHRR on performance, in addition to the FC results for the three settings of $h$. On certain datasets, CHRR outperformed FC even when it had vectors with lower dimensions. These results suggest that CHRR has a higher capacity for learning on datasets with a large number of labels than FC does.

We also compared CHRR with HRR. As shown in Figure 5, CHRR was better than HRR in many cases. In particular, the results for P@20 and PSP@20, where the value of the evaluation index k is large, we confirmed that the difference in performance is significant. As our theoretical experiment in § 3.2 showed, CHRR could represent many labels with high accuracy even for low-dimensional

---

[3]https://github.com/NeuromorphicComputationResearchProgram/
Learning-with-Holographic-Reduced-Representations
[4]For the Delicious-200K, we used the average scores of three trials and $h = \{768, 2048\}$ because training on the dataset requires a large computational time.

Table 3: Accuracy of our CHRR model and (FC) networks, and the left number in **bold** represents the compression ratio $\left(1 - \frac{(F \times h_C + h_C \times h_C) + (h_C \times 2d + d \times L)}{(F \times h_F + h_F \times h_F) + (h_F \times L)}\right)$ of the CHRR's model size for FC's model size. CHRR is set with $d = 800$ and $h_C = 768$. And the right number in **bold** represents the compression ratio $(1 - \frac{d}{L})$ of the CHRR's output dimensions for FC's output dimensions. For FC, $d$ is set at the number of labels in each dataset $(L)$ and $h$ is set at 2048.

| (d, h) | Delicious (**59**%, **19**%) | | EURLex-4K (**61**%, **80**%) | |
| | FC (983, 2048) | CHRR (800, 768) | FC (3993, 2048) | CHRR (800, 768) |
| --- | --- | --- | --- | --- |
| P@1 | 70.8($\pm$0.2) | 69.8($\pm$0.4) | 77.4($\pm$0.6) | 75.2($\pm$0.4) |
| P@5 | 59.2($\pm$0.0) | 59.0($\pm$0.3) | 47.9($\pm$0.2) | 47.8($\pm$0.1) |
| P@10 | 49.7($\pm$0.2) | 49.1($\pm$0.2) | 33.2($\pm$0.2) | 29.8($\pm$0.0) |
| PSP@1 | 34.1($\pm$0.2) | 33.3($\pm$0.3) | 33.6($\pm$0.5) | 28.7($\pm$0.2) |
| PSP@5 | 36.0($\pm$0.1) | 35.6($\pm$0.3) | 37.3($\pm$0.2) | 34.9($\pm$0.1) |
| PSP@10 | 36.1($\pm$0.2) | 35.6($\pm$0.3) | 51.2($\pm$0.4) | 42.3($\pm$0.1) |
| (d, h) | Wiki10-31K (**61**%, **99**%) | | Delicious-200K (**62**%, **99**%) | |
| | FC (101938, 2048) | CHRR (800, 768) | FC (205443, 2048) | CHRR (800, 768) |
| P@1 | 80.5($\pm$3.0) | 82.2($\pm$0.5) | 35.1($\pm$0.6) | 43.2($\pm$0.1) |
| P@5 | 46.3($\pm$2.2) | 58.8($\pm$0.3) | 32.1($\pm$0.4) | 37.1($\pm$0.1) |
| P@10 | 36.8($\pm$5.8) | 42.0($\pm$0.1) | 29.5($\pm$0.3) | 33.1($\pm$0.3) |
| PSP@1 | 10.5($\pm$1.5) | 10.2($\pm$0.0) | 5.3($\pm$0.1) | 6.6($\pm$0.1) |
| PSP@5 | 8.9($\pm$0.5) | 10.9($\pm$0.1) | 7.4($\pm$0.1) | 8.5($\pm$0.0) |
| PSP@10 | 10.4($\pm$3.0) | 11.2($\pm$0.1) | 8.8($\pm$0.1) | 9.9($\pm$0.1) |

vectors. The results of the theoretical experiments in § 3.2 and the experiment on real datasets in § 5 suggest that the CHRR is able to represent a larger number of correct labels. Moreover, the results of additional experiments reported in Appendix B show that CHRR also outperformed HRR on the datasets that we created with a large number of labels per sample.

## 5.5 IMPACT OF MODEL ARCHITECTURE

This section describes the results of the experiments on the impact of the model architectures in § 4.2. Figure 6 compares the performances of the CHRR variants (CHRR, CHRR-Half, CHRR-sin, and CHRR-tanh) on the Wiki10-31K dataset. As mentioned in § 4.2, there was no significant difference in performance between CHRR and this model. CHRR-sin and CHRR-tanh both obtained similar results that were inferior to those of CHRR and CHRR-Half. While the sin function in CHRR-sin seems to consider the cyclic feature, the results show that it is imperfect at predicting the of the circular-label vector. In short, our developed network architecture is important for the XMC learning with circular vectors, while the increase in the model size is not a big issue.

## 6 CONCLUSION

The XMC task still faces challenge of dealing with a large number of output labels. In this paper, we attempted to address this issue by using a low dimensional circular vector to output directly. In theoretical experiments in § 3.2 and § 3.3, we showed that many labels can be accurately encoded by using circular vectors (CHRR) rather than normal real-valued vectors (HRR). Moreover, using actual XMC datasets, we compared the accuracy with CHRR, HRR, and FC in § 5. CHRR reduced the output layer size by up to 99% compared to FC, while it outperformed FC in most results. Comparing HRR and CHRR, CHRR outperformed on most results. In addition, the larger the number of labels per sample in the data set, the larger the performance difference between CHRR and HRR became. As described above, our proposed circular vectors system contributed to a significant improvement in the XMC task. In the future, we will study the impact of the hidden layer size $h$ on accuracy. A further step is to incorporate circular vector systems into other DNN models such as

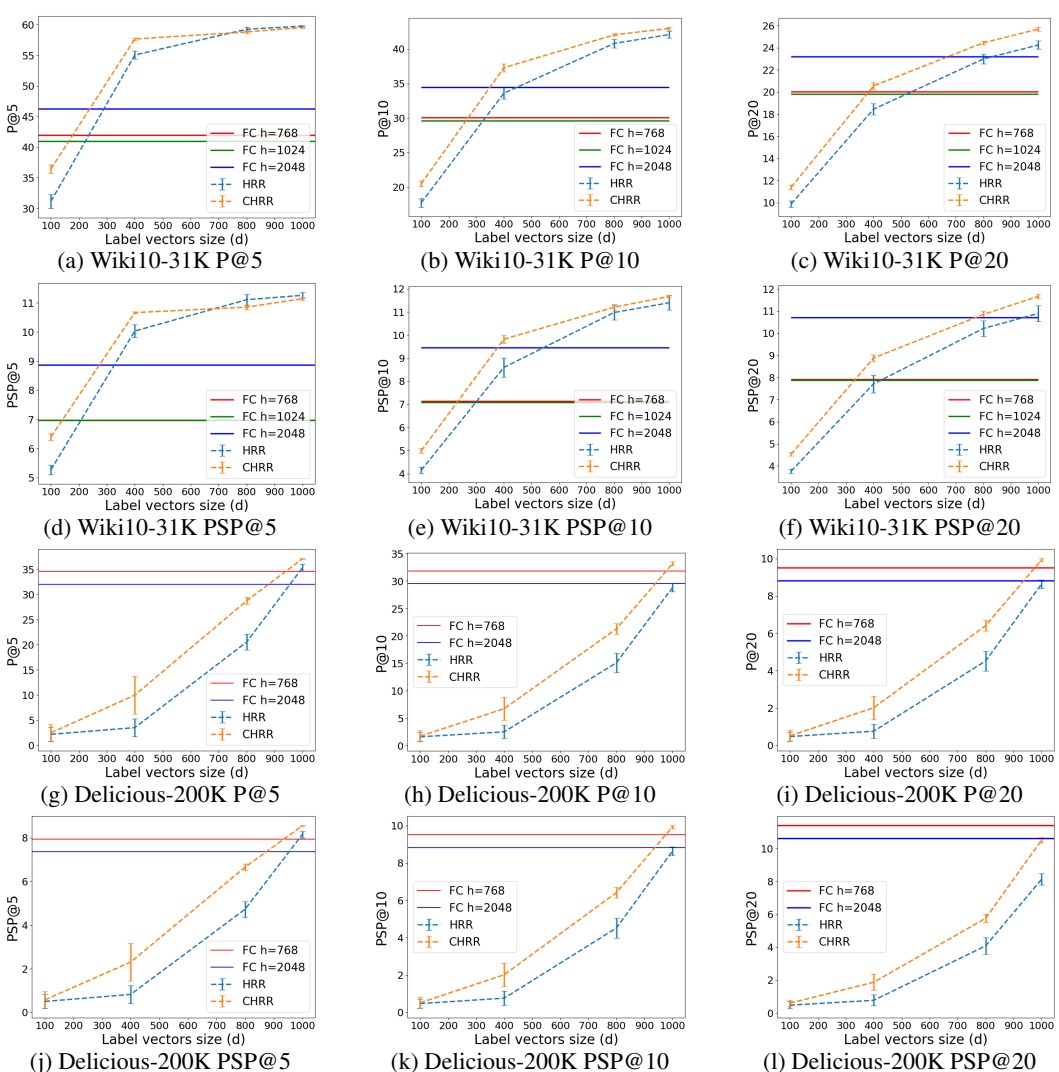

Figure 5: Impact of the number of dimensions ($d$) on P@5, P@10, P@20, PSP@5, PSP@10, and PSP@20 for Wiki10-31K and Delicious-200K datasets. The line of `FC h=768` in (d) overlaps with the line of `FC h=1024.`

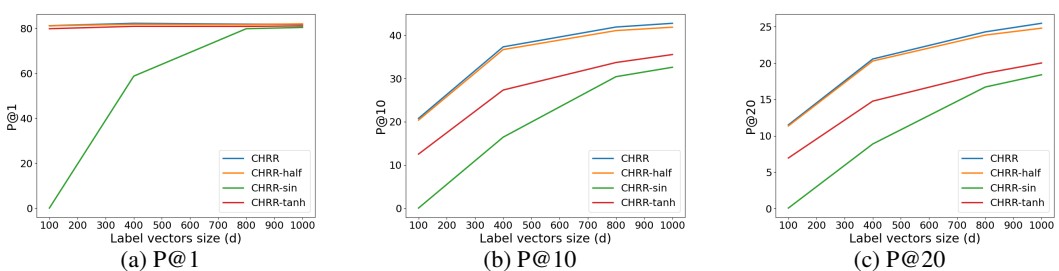

Figure 6: Comparison of the performances of CHRR variants (CHRR, CHRR-Half, CHRR-sin, and CHRR-tanh) on the Wiki10-31K dataset.

LSTM (Hochreiter & Schmidhuber, 1997), Transformer (Vaswani et al., 2017), as well as Associative LSTM (Danihelka et al., 2016) and Hrrformer (Alam et al., 2023) for read-valued vector.

ETHICS STATEMENT

We used the publicly available XMC datasets, Delicious, EURLex-4K, Wiki10-31K and Delicious-200K, to train and evaluate DNN models, and there is no ethical consideration.

REPRODUCIBILITY STATEMENT

As mentioned in § 5.3, we used the publicly available code to implement FC, HRR and CHRR. Our code will be available at `https://github.com/[innominated]`.

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

## A    APPENDIX: EXPERIMENT TO INVESTIGATE THE EFFECT OF THE NUMBER OF HIDDEN LAYERS

Additional experiments were conducted to verify the behavior of FC and CHRR when the number of hidden layers was increased to three. In FC, the number of hidden layers was considered in both 2-layer and 3-layer cases, and the dimensionality of the hidden layer $h$ was set to $\{768, 1024, 2048\}$. In the CHRR experiment, the number of hidden layers was set to three and $h$ was set to $2048$. Table 4 shows that the performance of FC improves with increasing $h$, but the number of hidden layers itself does not contribute significantly to performance. On the other hand, the CHRR was confirmed to be lower than the P@10 of $42.0$ for the case set with two 768-dimensional hidden layers in Table 3, even when the number of hidden layers was increased and h was increased.

Table 4: Comparison of accuracy with changes in hidden layers. $h$ is the number of hidden layer dimensions and $d$ is the number of output dimensions. The dataset used was Wiki10-31K.

| Net. | Layers | $h$ | $d$ | P@10 |
|------|--------|-----|-----|------|
| FC | 2 | 768 | 30,938 | $27.8(\pm1.0)$ |
| | 2 | 1,024 | 30,938 | $29.8(\pm3.0)$ |
| | 2 | 2,048 | 30,938 | $36.8(\pm5.8)$ |
| | 3 | 768 | 30,938 | $30.8(\pm1.7)$ |
| | 3 | 1,024 | 30,938 | $31.4(\pm4.2)$ |
| | 3 | 2,048 | 30,938 | $34.9(\pm0.5)$ |
| CHRR | 3 | 2,048 | 800 | $38.9(\pm0.2)$ |
| | 3 | 2,048 | 1,000 | $39.6(\pm0.2)$ |
| | 3 | 2,048 | 1,500 | $40.5(\pm0.4)$ |

# B   APPENDIX: EXPERIMENTS ON DATASETS WITH A LARGE NUMBER OF LABELS PER SAMPLE

In § 5, we discussed the potential of the CHRR for representing a larger number of correct labels. This appendix provides additional experimental details examining the capacity of CHRR, especially on datasets we created that have a large number of labels per sample. The experiments used the remaining training data remained and the testing set to include only the top 10% of data samples with more labels. We named these datasets Delicious top10, Wiki10-31K top10, and Delicious-200K top10. This alteration allowed us to place a particular focus on how the CHRR metric behaves when subjected to data samples teeming with labels.

## B.1   TEST DATA SELECTION

We evaluated our method on the three datasets with only the number of top 10% labels as test data. Table 5 shows the details of the datasets. These datasets have the maximum of the average number of labels per samples ($\hat{L}_{\max}$) is $20,471$ of Delicious-200K.

Table 5: Detailed datasets we created statistics from (Bhatia et al., 2016). Here, $L$ is the number of labels, $\hat{L}$ is the average number of labels per sample, $\hat{L}_{\min}$ is the minimum number of labels per sample, and $\hat{L}_{\max}$ is the maximum number of labels per sample.

| Dataset | $L$ | $\hat{L}$ | $\hat{L}_{\min}$ | $\hat{L}_{\max}$ |
|---|---|---|---|---|
| Delicious top10 | 983 | 24.39 | 24 | 25 |
| Wiki10-31K top10 | 101,938 | 28.19 | 28 | 30 |
| Delicious-200K top10 | 205,443 | 406.15 | 181 | 20,471 |

## B.2   RESULT AND DISCUSSION

Figure 7 shows that CHRR is also more effective than HRR for the datasets with a large numbers of labels per sample. This tendency is almost the same as that of the results shown in Figure 5.

# C   APPENDIX: EXPERIMENTS

We conducted additional experiments by using input features generated by XLNet [Chang et. al. 2020] instead of BoW features to investigate the potential of transformer-based models. The results were comparable with recent models such as LSTM. [5]

---

[5]LSTM results were taken from (Ganesan et al., 2021).

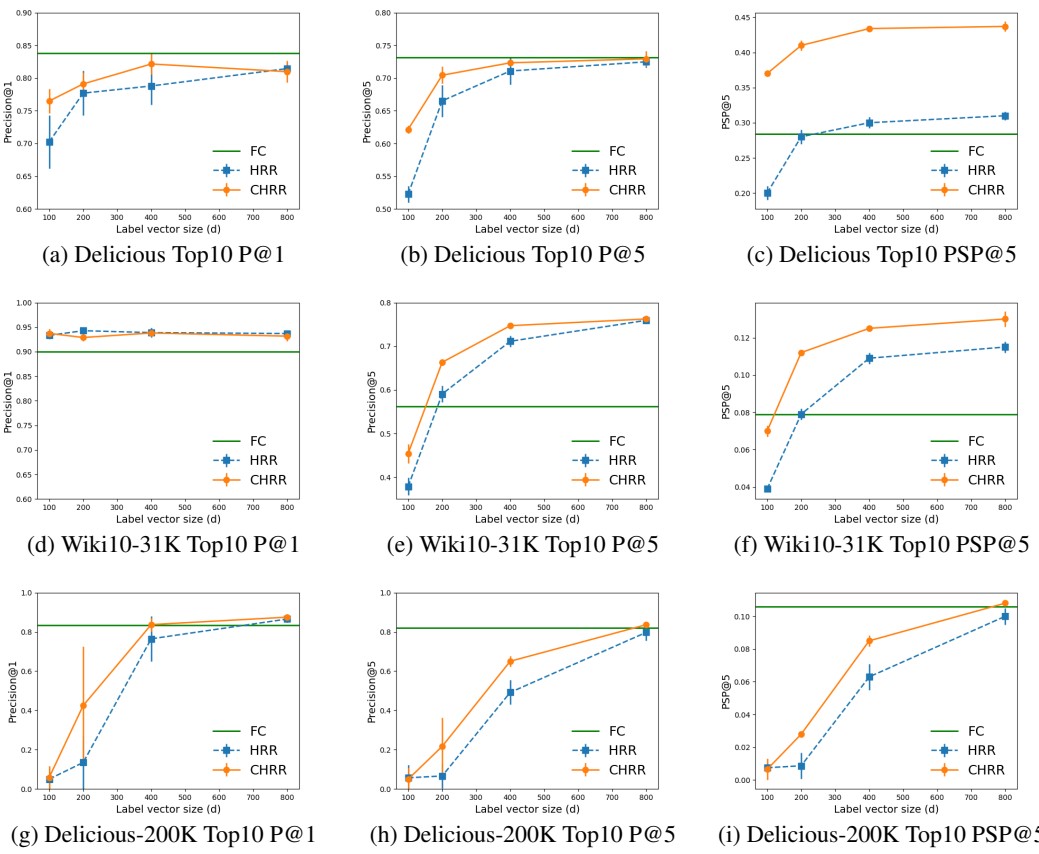

Figure 7: Impact of the number of dimensions ($d$) on P@1, P@5, and PSP@5 for Delicious Top10, Wiki10-31K Top10, and Delicious-200K Top10 datasets.

Table 6: Comparison of performance of FC, CHRR, FC-XLNet, CHRR-XLNet and LSTM, and the left number in **bold** represents the compression ratio $\left(1 - \frac{(F \times h_C + h_C \times h_C) + (h_C \times 2d + d \times L)}{(F \times h_F + h_F \times h_F) + (h_F \times L)}\right)$ of the CHRR's model size for FC's model size. And the right number in **bold** represents the compression ratio $(1 - \frac{d}{L})$ of the CHRR's output dimensions for FC's output dimensions. The dataset used was Wiki10-31K. *This LSTM score is reported by Ganesan et al. (2021).

| (d,h) | FC (101938,2048) | CHRR(**61**%, **99**%) (800,768) | FC-XLNet(**22**%, **0**%) (101938,2048) |
|---|---|---|---|
| P@1 | 80.5 | 82.2 | 84.0 |
| P@5 | 46.3 | 58.8 | 58.9 |
| P@10 | 36.8 | 42.0 | 43.6 |
| PSP@1 | 10.5 | 10.2 | 10.7 |
| PSP@5 | 8.9 | 10.9 | 11.0 |
| PSP@10 | 10.4 | 11.2 | 11.9 |
| | FC-XLNet(**−58**%, **0**%) (101938,4096) | CHRR-XLNet(**48**%, **99**%) (1500,768) | LSTM* - |
| P@1 | 85.1 | 86.1 | 83.5 |
| P@5 | 61.7 | 62.7 | - |
| P@10 | 46.1 | 45.8 | - |
| PSP@1 | 11.4 | 11.0 | - |
| PSP@5 | 12.2 | 12.0 | - |
| PSP@10 | 13.5 | 12.5 | - |

