# OpenReview forum: "Multi-label Learning with Random Circular Vectors"
_ICLR.cc/2024/Conference — Submitted to ICLR 2024_

### Official Review · Reviewer_Cc3g · 2023-10-23

**Soundness:** 3 good
**Presentation:** 3 good
**Contribution:** 3 good
**Rating:** 5
**Confidence:** 3

**Summary:**

This paper proposes a new strategy to mitigate the large computational and resource expenses of deep neural networks in the context of the extreme multi-label classification task. The authors advocate using random circular vectors as the prediction of the final layer of classifiers, where each vector component is represented as a complex amplitude. The authors claim that the proposed method helps to decrease the scale of output layers and improve performance. The provided experiments confirm this assertion.

**Strengths:**

- The authors proposed an approach to improve the ability to represent data instances that belong to many classes.

 - The authors provide extensive experiments to prove that the proposed approach outperforms its counterpart.

 - The proposed approach (CHRR) involves double output nodes in comparison with HRR, however, experiments show that by halving CHRR nodes into two groups, CHRR-Half is able to maintain similar performance while mitigating the scaling problem.

**Weaknesses:**

- The variances of the proposed model (CHRR-sin, CHRR-tanh) show minimal empirical improvement in the provided experiments. The motivation of it is also ambiguous. The authors are recommended to provide more details to explain or reconstruct this part.

 - Specifically, the absolute value of CHRR appears to be comparable between Figure 5(a)/6(a) and Figure 5(b)/6(b), but this consistency is not reflected in Figure 5(c)/6(c), even though they exhibit very similar trends. It is advised that the authors thoroughly review their figures and tables to eliminate any potential errors or misuses. If this is not the case, the authors are encouraged to provide a clear explanation of the notable performance improvement.

-  The authors give less-than-convincing explanation on the problem of `Wiki10-31K P@1` performance. The increase `Wiki10-31K P@5` and `Wiki10-31K P@10`  would not be as dramatic as it is if the explanation is valid. The authors are recommended to provide a convincing explanation of this problem.

- The authors underscore that the proposed methods reduce the size of the output layer, but few details are provided. On what criteria do the authors conclude that the output size is reduced by 59%~97%? What is the figure for each experiment?

- The details of network FC are not provided. The authors claim that they adopt the code provided by Ganesan et al 2021 but no further information is available.  The authors should provide a clear setting for it.

**Questions:**

- Can authors provide any insight on the performance degradation in Fig 7(a, b) other than section5.4?

---

> ### Author Response · Authors · 2023-11-15
> **Explanations for your concerns about our experiments**
>
> Reviewer Cc3g: We would like to express our sincere appreciation for your careful reading and valuable feedback on our paper. We greatly appreciate your insightful comments and questions.
>
> ### Weakness & Questions
>
> #### W1. The variances of the proposed model (CHRR-sin, CHRR-tanh) show minimal empirical improvement in the provided experiments. The motivation of it is also ambiguous. The authors are recommended to provide more details to explain or reconstruct this part.
>
> - A1. As you pointed out, the motivation for the experiment was not clearly stated. One of the main contribution is that the proposal of an effective method to integrate CHRR into a deep learning model. Therefore, we conducted the experiment to demonstrate the advantage of the proposed architecture (CHRR) against naive implementation (CHRR-sin, CHRR-tanh). We added the mention for the motivation into S4.2 to clarify it.
>
> #### W2. Specifically, the absolute value of CHRR appears to be comparable between Figure 5(a)/6(a) and Figure 5(b)/6(b), but this consistency is not reflected in Figure 5( c )/6( c ), even though they exhibit very similar trends. It is advised that the authors thoroughly review their figures and tables to eliminate any potential errors or misuses. If this is not the case, the authors are encouraged to provide a clear explanation of the notable performance improvement.
>
> - A2.
> - First, we are sorry for putting an incorrect figure in Figure 6( c ). In the current draft paper, we replaced all figures with correct ones.
> - For a detailed analysis, we conducted more experiments by varying the dimension size of hidden layers of FC, HRR and CHRR from 768 to 2,048, respectively. We displayed the results in Figure 5 and in Table 3 of our new draft paper. We think that the results clearly show the advantage of CHRR over HRR. If the problem still remains, it would be helpful if you could point it out again.
>
> #### W3. The authors give less-than-convincing explanation on the problem of Wiki10-31K P@1 performance. The increase Wiki10-31K P@5 and Wiki10-31K P@10 would not be as dramatic as it is if the explanation is valid. The authors are recommended to provide a convincing explanation of this problem.
>
> - A3. Since P@5,10,20 are more important than P@1 for XMC, Figure 5 in our current draft paper shows the results for P@5 instead of P@1.
>
> #### W4. The authors underscore that the proposed methods reduce the size of the output layer, but few details are provided. On what criteria do the authors conclude that the output size is reduced by 59%~97%? What is the figure for each experiment?
> - A4.
> - In our current draft paper, we reported the compression rate ((1-d/L) * 100%) (d: dimension size of output layer, L: the number of labels) and did not include the size "d * L" of non-learnable label matrix of CHRR/HRR. We will consider to show the model size (h * L for FC and h * d + d * L for CHRR/HRR) in the final version of our paper. We would like to note that when h=2,048 for FC, the model size of CHRR/HRR (h=768) is smaller than that of FC.
>
>
> #### W5. The details of network FC are not provided. The authors claim that they adopt the code provided by Ganesan et al 2021 but no further information is available. The authors should provide a clear setting for it.
>
> - A5. We are sorry for omitting the details. In S5.3 of our current draft paper, we described the information about the FC network architectures used in our experiments.
>
> #### Q1. Can authors provide any insight on the performance degradation in Fig 7(a, b) other than section5.4?
>
> - A6. Data samples belonging to many class labels represent a variety of concepts and it may be difficult to make label predictions for such data samples.
>
> ---

---

> > ### Comment · Reviewer_Cc3g · 2023-11-22
> >
> > I appreciate the authors for their comprehensive responses. There still remain some problems:
> >
> > 1. There seems to be a red line missing in Figure 5(d)
> > 2. I agree with the author that the model size, including the non-learnable label matrix should be indicated in Table3
> > 3. I share the concerns with other reviewers on the backbone. The authors should provide a comparison on different models other than FC.
> >
> > The authors have not provided their final paper yet. Thus, I keep my score.

---

> > > ### Author Response · Authors · 2023-11-22
> > > **Appreciating your response**
> > >
> > > We would like to thank you for your devoted time in reading our responses carefully. We updated our draft paper again.
> > >
> > > #### There seems to be a red line missing in Figure 5(d)
> > > - This is due to the overlapping of two lines (h=768 and 1024). We added the mention to the caption of Figure 5.
> > >
> > > #### I agree with the author that the model size, including the non-learnable label matrix should be indicated in Table3
> > > - We added the information about the "model" compression ratio to Table 3. Though CHRR includes the parameters of the non-learnable label matrix, it was still much smaller than FC for all datasets.
> > >
> > > #### The authors should provide a comparison on different models other than FC.
> > > - We put Table 6 in Appedix C, where we compared CHRR w/XLNet with FC w/XLNet and LSTM. We also note that the results of CHRR w/XLNet were comparable with current models.

---

### Official Review · Reviewer_7H1H · 2023-10-29

**Soundness:** 2 fair
**Presentation:** 3 good
**Contribution:** 2 fair
**Rating:** 5
**Confidence:** 3

**Summary:**

Existing DNN methods for XMC problem often consist of a large output label matrix where each column corresponds to a trainable label vector. This paper proposes the use of random circular vectors as non-learnable label vectors, which significantly reduce the trainable model parameters. The author proposed Circular-HRR (CHRR), which represents random circular vectors in complex domain, and designs a model architecture that predicts a low-dimensional circular vector. On moderate-size XMC datasets, the proposed CHRR method performs better than the fully-connected baseline as well as the previous method HRR.

**Strengths:**

1. The overall presentation of the paper is clear and easy to follow

2. Using circular vectors with complex amplitude is technically sound

**Weaknesses:**

1. CHRR do not reduce model parameters at the inference stage, compared to the FC baseline.

2. The inference time complexity of CHRR is as high as `O(L)`  while FC and HRR can be `O(log(L))`.

3. The experiment results are not very comprehensive. see detailed questions below.

**Questions:**

1. The proposed method (CHRR) reduces the "trainable" model parameters by using a non-learnable label matrix at the training stage. However, CHRR do not reduce the model parameters at inference stage, because it still need to store the non-learnable label matrix for finding top-k most relevant labels, given the model-predicted output random circular vector. What's the model parameters used at inference time, compared to HRR and FC?

2. At the inference stage, to compute similarity between the label matrix and the model predicted random circular vector, CHRR seems to involve more complex computations (Table 1) that is non-standard euclidean distance metric. This suggests CHRR can not enjoy the advantage of fast/approximate nearest neighbor search methods that reduces the inference time complexity from `O(L)` to `O(log(L))` where `L` is the number of labels. On the other hand, HRR can actually leverage ANN search methods at the inference stage. Any analysis on the inference time complexity and the actually inference latency?

3. The experiment results are not very comprehensive:
(1) There should be a table comparing the model size at training/inference stage, and the detail hyper-parameters such as `h` and `d`.
(2) Compared to the FC baseline, It is not clear whether the performance gain of CHRR is from larger model size (including the non-learnable label matrix).
(3) The proposed method did not compare with more advanced XMC methods, such as Transformer-based encoders.

---

> ### Author Response · Authors · 2023-11-15
> **Explanations for your concerns about our experiments and model efficiency**
>
> Reviewer 7H1H: We would like to express our gratitude for your valuable feedback on our paper. We greatly appreciate your insightful comments and questions.
>
> ### Weakness & Questions
>
> #### W1. CHRR do not reduce model parameters at the inference stage, compared to the FC baseline.
> #### Q1. What's the model parameters used at inference time, compared to HRR and FC?
> #### W2. The inference time complexity of CHRR is as high as O(L) while FC and HRR can be O(log(L)).
> #### Q2. Any analysis on the inference time complexity and the actually inference latency?
>
> - A1.
> - First, we could improve the inference efficiencies of both HRR and CHRR using an Approximate Nearest Neighbor (ANN) method for the Maximum Inner Product Search (MIPS). The reason why we can apply MIPS to circular vectors is that the CHRR’s similarity operation between two circular vectors θ and φ is actually equivalent to the inner product between two real-valued vectors, which are obtained via the inverse FFTs of  θ and φ, respectively (The fact is simply based on the Parseval’s theorem).
> - As you pointed out, HRR and CHRR require d * L parameters for keeping non-learnable label matrix in the inference stage.
> - Speedups in training time for CHRR against FC were almost the same as those for HRR reported in Table 3 of [Ganesan et. al. 2021], while CHRR provides a clean and better label encoding/retrieval framework for XMC than HRR. While CHRR does not require any FFT/iFFT in the inference stage, its similarity computation contains the calculation of the cos function, which makes the inference slow. From our experimental results, the computational cost for CHRR is almost the same as that of HRR in practice.
>
> #### W3. The experiment results are not very comprehensive. see detailed questions below.
>
> #### Q3.(1) There should be a table comparing the model size at training/inference stage, and the detail hyper-parameters such as h and d.
> - A21.
> - Please see Table 3 of our new draft paper.
> - For a detailed analysis, we conducted more experiments by varying the dimension size of hidden layers of FC, HRR and CHRR from 768 to 2,048, respectively. Especially, the performance of FC was improved when h=2,048. We displayed the new results in Figure 5 and Table 3 of our new draft paper. In Table 3, we described the compression rate ((1-d/L) * 100%), but as you pointed out, we will consider to show the model size (h * L for FC and h * d + d * L for CHRR) in the final version of our paper.
>
> #### Q3.(2) Compared to the FC baseline, it is not clear whether the performance gain of CHRR is from larger model size (including the non-learnable label matrix).
> - A22.
> - According to A21, we confirmed that the performance gain of CHRR is not necessarily caused by a larger model size. For example, to see Figure 5 and Table 3, CHRR (h=768) with d>=400 on Wiki10-31K outperforms FC in all metrics while the model size of CHRR (including the non-learnable label matrix) is smaller than FC.
> - We conducted additional experiments with a larger dimension size (h=2,048) of hidden layers in FC, where the number of model parameters was larger than that of CHRR/HRR. However, CHRR/HRR still outperformed FC for several datasets. Though the reason is unclear, we suspect that FC requires a larger dimension size for hidden layers due to an extremely large output space. We would like to further investigate this point in the final version (since additional experiments require a lot of machine resources.).
>
> #### Q3.(3) The proposed method did not compare with more advanced XMC methods, such as Transformer-based encoders.
> - A23.
> - For Wiki10-31k, we conducted additional experiments by using input features generated by XLNet [Chang et. al. 2020] instead of BoW features to investigate the potential of transformer-based models. The results were comparable with recent models.
>     -   | FC w/XLNet |h=768|h=1024|h=2048|h=4096|
>         | ----       | --- | ---  | ---  | ---  |
>         | P@1        |81.5 |83.3  |84.0  |85.1  |
>         | P@5        |54.2 |56.6  |58.9  |61.7  |
>         | P@10       |39.2 |41.6  |43.6  |46.1  |
>         | P@20       |26.2 |27.9  |29.5  |31.2  |
>
>     -   | CHRR w/XLNet |h=768,d=800|h=768,d=1500 |
>         | ---- | --- | --- |
>         | P@1  |85.5     |86.1     |
>         | P@5  |61.5     |62.7     |
>         | P@10 |43.7     |45.8     |
>         | P@20 |25.5     |28.8     |
> - In the final version of this paper, we will show more experimental results using fine-tuned LMs for other datasets including AmazonCat-13K and Wiki-500K. Moreover, we will try a combination of XLNet and TF-IDF features for further performance improvements as X-Transformer [Chang et. al. 2020] has done.
>
> ---

---

> > ### Author Response · Authors · 2023-11-22
> > **Additional modification for Table 3**
> >
> > ## Additional modification for Table 3
> > #### W1 and Q3(2)
> > - We added the information about the "model" compression ratio to Table 3. Though CHRR includes the parameters of the non-learnable label matrix, it was still much smaller than FC for all datasets.

---

### Official Review · Reviewer_vWH8 · 2023-10-30

**Soundness:** 2 fair
**Presentation:** 3 good
**Contribution:** 1 poor
**Rating:** 5
**Confidence:** 4

**Summary:**

This paper is about improving the representative power of embeddings in Extreme Multi-label Learning (XML) with ideas from Holographic Reduced Representations (HRR). Typical XML approaches, which leverage a large linear classifier layer to map an input to a label set, have limitations: the linearity constraint restricts the modelling power while the enormous number of classifiers blow-up the time and space complexities. Several approaches have been proposed to mitigate the complexity issues, e.g. tree-search based and negative label mining based approaches. This paper alternatively proposes to use HRRs which learn classifiers in the Fourier-transformed space instead of the original linear embedding space, thus resulting in more powerful, non-linear classifier learning. The proposed approach also brings down the training complexity by leveraging loss functions that depend only on the positive labels (which are typically sparse).

The paper largely exploits the key ideas from earlier papers [HRR in Plate '95, HRR for XML in Ganesan et.al. '21]. In addition, it generalizes  [Ganesan et.al.'21] through Complex-valued HRR representations. Experiments demonstrate that, keeping other factors constant, the accuracy with CHRR > HRR > naive fully connected XML classifier layer.

**Strengths:**

* The paper introduces the innovative idea of complex-valued holographic reduced representations (CHRR) for XML tasks which can significantly improve the XML prediction accuracy over and above that achieved by real-valued HRR

* Experiments demonstrate the efficacy of proposed approach on several moderately large-scaled XML datasets in terms of P and PSP gains

**Weaknesses:**

* The contributions of this paper are rather limited. The key ideas behind adapting HRR to XML, such as unitary normalization and HRR XML loss, have been borrowed from [Ganesan et.al. '21]. The main novelty lies in generalizing real to complex HRR. While this is useful, its efficiency-accuracy trade-offs relative to original HRR have not been well established.

* The experimental validation of proposed approach is weak.
- Datasets involve moderate scale XML datasets and none from >1 million scale
- Datasets considered are bag-of-words based whereas contemporary XML literature has shifted focus to transformer-learnt representations
- Model architecture considered is based on fully-connected layers whereas contemporary XML literature has shifted focus to transformers
- No comparison is provided with existing schemes to reduce training complexity such as tree, hashing or negative label sampling based approaches
- Even though the main claim of this paper is to reduce training complexity, no training time comparisons have been reported
- Due to all these factors, the real utility and impact towards XML field due to this paper is hard to evaluate. A substantial amount of additional work is needed in this direction

* The proposed approach only improves training time and does not appear to reduce prediction time or memory requirements which are also important requirements in XML

**Questions:**

* What are the cost-accuracy trade-offs of CHRR vs HRR with cost measured in wallclock time and ram requirements?
* How does CHRR fare relative to FC and HRR on BERT based architectures and on datasets with much larger quantity of labels?
* What are the relative advantages and disadvantages of CHRR versus tree, hashing or negative label sampling based approaches to reduce XML training and prediction complexities ?

---

> ### Author Response · Authors · 2023-11-15
> **Novelty of our work (Weakness)**
>
> Reviewer vWH8: We would like to express our gratitude for your valuable feedback on our paper. We greatly appreciate your insightful comments and questions.
>
> ### Weakness
>
> #### W1. The contributions of this paper are rather limited.
>
> - A1. We would like to highlight that our main contributions are; (1) We revealed the issue on the unitary normalization method of [Ganesan et.al. '21] in S3 and (2) We proposed an effective method to integrate CHRR into a Deep Learning (DL) framework as presented in S4. In the point (1), we raised a new perspective rather than borrowing ideas from [Ganesan et.al. '21]. In the second point (2), it was not obvious how to integrate CHRR into a DL framework as we discussed in S4.2 and S5.5. We believe that these two points are our concrete contributions.
>
> #### W2. The experimental validation of proposed approach is weak.
>
> #### W3. Datasets involve moderate scale XML datasets and none from >1 million scale
>
> #### W4. Datasets considered are bag-of-words based whereas contemporary XML literature has shifted focus to transformer-learnt representations
>
> - A2.
> - For Wiki10-31k, we conducted additional experiments by using input features generated by XLNet [Chang et. al. 2020] instead of BoW features to investigate the potential of transformer-based models. The results were comparable with those of recent models.
>     -   | FC w/XLNet |h=768|h=1024|h=2048|h=4096|
>         | ----       | --- | ---  | ---  | ---  |
>         | P@1        |81.5 |83.3  |84.0  |85.1  |
>         | P@5        |54.2 |56.6  |58.9  |61.7  |
>         | P@10       |39.2 |41.6  |43.6  |46.1  |
>         | P@20       |26.2 |27.9  |29.5  |31.2  |
>
>     -   | CHRR w/XLNet |h=768,d=800|h=768,d=1500 |
>         | ---- | --- | --- |
>         | P@1  |85.5     |86.1     |
>         | P@5  |61.5     |62.7     |
>         | P@10 |43.7     |45.8     |
>         | P@20 |25.5     |28.8     |
> - In the final version of this paper, we will show more experimental results using fine-tuned LMs for other datasets including AmazonCat-13K and Wiki-500K. Moreover, we will try a combination of XLNet and TF-IDF features for further performance improvements as X-Transformer [Chang et. al. 2020] has done.
> - For a detailed analysis, we also conducted more experiments by varying the dimension size of hidden layers of FC, HRR and CHRR from 768 to 2,048, respectively. We displayed the new results in Figure 5 and Table 3 of our new draft paper. Especially, for Delicious-200k, CHRR outperformed HRR more clearly, compared to the older version. We here used P@5 instead of P@1 for clear explanations (We think that P@k (k>=5) is much important than P@1 for XMC).
>
> #### W5. Model architecture considered is based on fully-connected layers whereas contemporary XML literature has shifted focus to transformers
> - A3.
> - Transformer-based models for XMC tasks (e.g. [Chang et. al. 2020]) employ some tricks such as label clustering to reduce the learning/prediction cost of the large language models. Such tricks make a fair comparison of the label representations of CHRR and HRR difficult. Therefore, we decided to use a fully-connected layer as our fundamental architecture.
> - On the other hand, we agree with your point. However, it is not easy to find a method to apply CHRR to complicated transformer-based models. Instead, we conducted additional experiments to investigate the potential of transformer-based models, as we mentioned in A2. In the future, we would like to investigate a better way to combine CHRR with transformer-based models.
>
> #### W6. No comparison is provided with existing schemes to reduce training complexity such as tree, hashing or negative label sampling based approaches
>
> - A4. The vector symbolic architectures like HRR are actually similar to the concept of Bloom Filter as shown in the following theoretical paper. We also think that comparisons with other methods like tree and hashing are important. In a furture work, we would like to investigate how to use the Autoscaling Bloom Filter (a hashing method) for training XMC models more efficiently.
>     - Kleyko et. al., "Autoscaling Bloom Filter: Controlling Trade-off Between True and False Positives", Neural Computing and Applications, 2019.
>
> #### W7. Even though the main claim of this paper is to reduce training complexity, no training time comparisons have been reported
>
> - A5. Speedups in training time for CHRR against FC were almost the same as those for HRR reported in Table 3 of [Ganesan et. al. 2021], while CHRR provides a clear and better label enconding/retrieval framework for XMC than HRR.

---

> ### Author Response · Authors · 2023-11-15
> **Novelty of our work (Questions)**
>
> (Continuing the Response to Reviewer vWH8)
>
> #### W8. The proposed approach only improves training time and does not appear to reduce prediction time or memory requirements which are also important requirements in XML
>
> - A6. We could improve the inference efficiencies of both HRR and CHRR using an Approximate Nearest Neighbor (ANN) method for the Maximum Inner Product Search (MIPS). The reason why we can apply MIPS to circular vectors is that the CHRR’s similarity operation between two circular vectors θ and φ is actually equivalent to the inner product between two real-valued vectors, which are obtained via the inverse FFTs of θ and φ, respectively (The fact is simply based on the Parseval’s theorem).
> ### Questions
>
> #### Q1. What are the cost-accuracy trade-offs of CHRR vs HRR with cost measured in wallclock time and ram requirements?
>
> - A7.
> - Regarding to the memory cost, CHRR has no clear disadvantage against HRR: CHRR has twice the label encoding capacity and can retrieve labels more accurately than HRR as we shown in S3.
> - However, while CHRR does not require FFTs and inverse FFTs, the similarity operation of CHRR requires the cos function, which makes the inference slow in practice. However, as we mentioned in A6, since the similarity operation of CHRR can be equivalently transformed into the inner product in the real-valued vector space, ANN methods could be used to reduce the computational complexity.
> - As we mentioned in A5, training time of CHRR was almost the same as HRR for all datasets.
>
> #### Q2. How does CHRR fare relative to FC and HRR on BERT based architectures and on datasets with much larger quantity of labels?
>
> - A8.
> - Each sample in Delicious-200k has several tens of labels on average, and CHRR outperformed HRR clearly on the dataset, as we shown in Figure 5 (g)-(l) of our new draft paper.
> - While there is no fine-tuned pretrained language model for Delicious-200k, FC, HRR and CHRR with a fine-tuned XLNet for Wiki10-31k achieved the following performances, which are comparable with recent Transformer-based models.
>     -   | FC w/XLNet |h=768|h=1024|h=2048|h=4096|
>         | ----       | --- | ---  | ---  | ---  |
>         | P@1        |81.5 |83.3  |84.0  |85.1  |
>         | P@5        |54.2 |56.6  |58.9  |61.7  |
>         | P@10       |39.2 |41.6  |43.6  |46.1  |
>         | P@20       |26.2 |27.9  |29.5  |31.2  |
>
>     -   | CHRR w/XLNet |h=768,d=800|h=768,d=1500 |
>         | ---- | --- | --- |
>         | P@1  |85.5     |86.1     |
>         | P@5  |61.5     |62.7     |
>         | P@10 |43.7     |45.8     |
>         | P@20 |25.5     |28.8     |
>
> #### Q3. What are the relative advantages and disadvantages of CHRR versus tree, hashing or negative label sampling based approaches to reduce XML training and prediction complexities ?
>
> - A9.
> - As we mentioned in A4, in our future work, we would like to compare CHRR with a hashing method based on Autoscaling Bloom Filter.
> - As we mentioned in A5, CHRR can improve the training efficiency of FC networks, while providing a better label encoding/retrieval framework for XMC than HRR.

---

> > ### Comment · Reviewer_vWH8 · 2023-11-22
> >
> > I thank the authors for addressing the concerns of novelty and for additional experiments with XLNet. However, I feel that, while the ideas presented in this paper are novel and potentially useful for XML, there is significant scope for improving the experimental validation to realize this potential. I would encourage the authors to revise their paper by improving the motivation, empirical results and discussion aspects of the paper and resubmit. The most important question for me is - Do these proposed ideas change the way XML is approached at present in any significant sense?
> >
> > As of now, I keep my score.

---

> > > ### Author Response · Authors · 2023-11-22
> > > **Contributions to the XMC community**
> > >
> > > We added the information about the “model” compression ratio to Table 3. Though CHRR includes the parameters of the non-learnable label matrix, it was still much smaller than FC for all datasets. We think that this is also a not small contribution to the XMC community since FC network (with a fine-tuned text encoder model) is simple but can be a strong baseline model for XMC, as we showed in Table 6 of Appendix C.

---

> ### Author Response · Authors · 2023-11-22
> **Appreciating your response**
>
> We would like to thank you for your devoted time in reading our responses carefully.
> We insist again that as demonstrated in Ganesan21 (Table 3), HRR-based approach for XMC is clearly effective in practice.
> As you mentioned, moreover, CHRR has several clear advantages over HRR.
> In this sense, we believe that CHRR is novel and potentially useful for XMC.

---

### Official Review · Reviewer_7cd9 · 2023-11-02

**Soundness:** 3 good
**Presentation:** 3 good
**Contribution:** 2 fair
**Rating:** 6
**Confidence:** 2

**Summary:**

This paper solves the issue by exploring the use of random circular vectors, where each vector component is represented as a complex amplitude. Specifically, the paper developed an output layer and loss function of DNNs for XMC by representing the final output layer as a fully connected layer that directly predicts a low-dimensional circular vector encoding a set of labels for a data instance.  Extensive experiments on synthetic datasets to verify the effectiveness of circular vectors.

**Strengths:**

1. The motivation is clear and the algorithm is sensible.
2. The proposed method is tested on several benchmarks.

**Weaknesses:**

The paper is in general easy to follow and well-structured. There are some interesting theoretical guarantees, which seem simple and effective. Nevertheless, I have the following concerns:

1. Not enough empirical evaluations. it necessary to evaluate other state-of-the-art benchmarks.
2. What is the computational cost of method? [addressed by rebuttal]
3.  Will the code be shared? [addressed by rebuttal].

**Questions:**

I am very impressed by the ideas and the writing of this paper. The method is simple and well-motivated. The evaluations address many aspects of the method.

---

> ### Author Response · Authors · 2023-11-15
> **Appreciating your attention: Explanations for your concerns about our experiments**
>
> Reviewer 7cd9: We greatly appreciate your encouraging comments and insightful questions.
>
> ### Weakness
>
> #### W1. Not enough empirical evaluations. it necessary to evaluate other state-of-the-art benchmarks.
> - A1.
> - For Wiki10-31k, we conducted additional experiments by using input features generated by XLNet [Chang et. al. 2020] instead of BoW features to investigate the potential of transformer-based models. The results were comparable with recent models as follows.
>     -   | FC w/XLNet |h=768|h=1024|h=2048|h=4096|
>         | ----       | --- | ---  | ---  | ---  |
>         | P@1        |81.5 |83.3  |84.0  |85.1  |
>         | P@5        |54.2 |56.6  |58.9  |61.7  |
>         | P@10       |39.2 |41.6  |43.6  |46.1  |
>         | P@20       |26.2 |27.9  |29.5  |31.2  |
>
>     -   | CHRR w/XLNet |h=768,d=800|h=768,d=1500 |
>         | ---- | --- | --- |
>         | P@1  |85.5     |86.1     |
>         | P@5  |61.5     |62.7     |
>         | P@10 |43.7     |45.8     |
>         | P@20 |25.5     |28.8     |
> - In the final version of this paper, we will show more experimental results using fine-tuned LMs for other datasets including AmazonCat-13K and Wiki-500K. Moreover, we will try a combination of XLNet and TF-IDF features for further improvements as X-Transformer [Chang et. al. 2020] has done.
> - For a detailed analysis, we conducted more experiments by varying the dimension size of hidden layers of FC, HRR and CHRR from 768 to 2,048, respectively. We displayed the results in Figure 5 and in Table 3 of our new draft paper. Especially, for Delicious-200k, CHRR outperformed HRR more clearly, and we think that the results support the main claim of this paper.
>
> #### W2. What is the computational cost of method? [addressed by rebuttal]
> - A2.
> - Speedups in training time for CHRR against FC were almost the same as those for HRR reported in Table 3 of [Ganesan et. al. 2021], while CHRR provides a clean and better label encoding/retrieval framework for XMC than HRR.
> - We could improve the CHRR’s inference for finding top-k most relevant labels using an Approximate Nearest Neighbor  (ANN) method for the Maximum Inner Product Search because the CHRR’s similarity operation between two circular vectors θ and φ is actually equivalent to the inner product between two real-valued vectors, which are obtained via the inverse FFTs of θ and φ, respectively (The fact is simply based on the Parseval’s theorem).
>
> #### W3. Will the code be shared? [addressed by rebuttal]
> - A3. Yes, we will make our codes publicly available from the first author’s github page.
>
> ---

---

> > ### Comment · Reviewer_7cd9 · 2023-11-22
> >
> > I have read the feedback from the authors and I still have concerns about insufficient experiments as well as presentation issues. I would encourage the authors to revise their paper by improving the discussion aspects of the paper. Thus, I maintain my review score.

---

> > > ### Author Response · Authors · 2023-11-22
> > > **Appreciating your response**
> > >
> > > We would like to thank you for your devoted time in reading our responses carefully.
> > > We agree with you, but we are very happy if you can tell us specifically which parts of our paper’s discussion need improvements.
> > > Though we updated our draft paper to make our contributions more clear, we would like to further improve its content if there are any more problems in it.

---

> > > ### Author Response · Authors · 2023-11-22
> > > **Additional information to Table3**
> > >
> > > We would like to tell you the following additional fact.
> > > We added the information about the “model” compression ratio to Table 3. Though CHRR includes the parameters of the non-learnable label matrix, it was still much smaller than FC for all datasets. We think that this is also a not small contribution to the XMC community since FC network (with a fine-tuned text encoder model) is simple but can be a strong baseline model for XMC, as we showed in Table 6 of Appendix C.

---

### Author Response · Authors · 2023-11-21
**Gentle reminder for the rebuttal**

(We are writing to gently remind you of the upcoming deadline for the rebuttal discussion period, which is set for the end of **November 22nd**.)

We would like to thank all reviewers for their valuable insights and comments made to improve this work. We updated our draft paper based on your valuable comments.
Moreover, please refer to the following additional experiments conducted to address your concerns (the experimental results will be reflected to the future version of our draft).

#### Concerns about comparisons with current models:
- The additional experimental results show that our method is comparable to state-of-the-art (SoTA) models, and it has clear advantages over the FC approach. These details will be included in the appendix of the final version.
    -   | FC w/XLNet |h=768|h=1024|h=2048|h=4096|
        | ----       | --- | ---  | ---  | ---  |
        | P@1        |81.5 |83.3  |84.0  |85.1  |
        | P@5        |54.2 |56.6  |58.9  |61.7  |
        | P@10       |39.2 |41.6  |43.6  |46.1  |
        | P@20       |26.2 |27.9  |29.5  |31.2  |

    -   | CHRR w/XLNet |h=768,d=800|h=768,d=1500 |
        | ---- | --- | --- |
        | P@1  |85.5     |86.1     |
        | P@5  |61.5     |62.7     |
        | P@10 |43.7     |45.8     |
        | P@20 |25.5     |28.8     |

#### Concerns about similarity computation time for CHRR:
- There was no critical difference between HRR and CHRR.
    -   |       |d=100|d=400 |d=800 |d=1,000|
        | ----  | --- | ---  | ---  | ---  |
        | HRR   |0.052 (s)|0.084 (s)|0.140 (s)|0.167 (s)|
        | CHRR  |0.051 (s)|0.099 (s)|0.165 (s)|0.200 (s)|
    - In this experiment, we used a lookup table for the computation of cosines in CHRR. The slow similarity calculation of CHRR can be caused by d cosines. However, the cosine computations could be relieved by using the lookup table.

---

### Meta-Review · Area_Chair_rY1Q · 2023-12-09

**Metareview:**

This paper introduces a coding scheme to improve the efficacy of learning in the context of extreme multi-label classification. Drawing on a previous study that used holographic reduced representations (HRR), the authors enhanced this approach by incorporating circular vectors to code multiple labels through angles. Empirically, the authors demonstrate that this method outperforms HRR and can be considered as an alternative for FC layers. Reviewers highlighted a common concern: the lack of comparison with state-of-the-art benchmarks. The responses provided during the reviewer-author discussion period did not result in increased scores, leading to ratings of 5, 5, 5, and 6. The AC agrees with the reviewers on the merit of the idea but notes a lack of clarity on how the method compares against current benchmarks. As such, the AC, with regret, votes for rejection.

**Justification For Why Not Higher Score:**

The proposed method (encoding via angles) is not fully justified, even empirically (the paper closely followed the HRR experiments). Given the fact that none of the reviewers championed the paper or even gave it a reasonably high score, I suggest rejecting this paper.

**Justification For Why Not Lower Score:**

NA

---

### Decision · Program_Chairs · 2024-01-16

Reject